# OpenReview forum: "MedFuzz: Exploring the Robustness of Large Language Models in Medical Question Answering"
_ICLR.cc/2025/Conference — Submitted to ICLR 2025_

### Official Review · Reviewer_6sJS · 2024-10-29

**Soundness:** 3
**Presentation:** 3
**Contribution:** 4
**Rating:** 5
**Confidence:** 5

**Summary:**

This paper proposes MedFuzz, a novel approach designed to evaluate the robustness of large language models (LLMs) in medical question-answering contexts. MedFuzz introduces controlled perturbations in input text by adding patient characteristics (PC) and social bias information to simulate real-world variability and challenges encountered in clinical settings.

The authors highlight the limitations of traditional medical benchmarks that often simplify clinical scenarios and position MedFuzz as an advancement towards “beyond-the-benchmark” evaluations. Specifically, the paper presents experiments assessing LLMs' responses to MedFuzz perturbations and evaluates the consistency of chain-of-thought (CoT) explanations under these conditions. The study offers a new perspective on testing LLM robustness by addressing potential risks in clinical decision-making when assumptions of canonical benchmarks do not hold.

**Strengths:**

1. This paper introduces MedFuzz, a novel approach for testing the robustness of large language models (LLMs) in clinical contexts, which addresses the simplifications found in traditional benchmarks. MedFuzz is distinct in its approach by adding specific patient characteristics and social bias information to simulate the complexity of real-world clinical scenarios. This innovative framework offers a new direction for assessing LLM robustness by examining potential vulnerabilities in medical question-answering settings.

2. The paper clearly explains the concept of MedFuzz and its application, particularly in using patient characteristics and bias elements to test model robustness. The experimental procedures and components are consistently described, making the study's objectives and methodology easy for readers to follow.

3. MedFuzz presents a significant contribution as it provides a framework to evaluate how LLMs may perform in real clinical settings, beyond simplified benchmarks. This work has high practical relevance for the safe implementation of LLMs in healthcare by strengthening robustness assessment and reducing potential errors. It contributes an essential tool for enhancing LLM applicability in clinical practice, highlighting the importance of robustness in medical AI.

**Weaknesses:**

1. The authors clarified the distinction between robustness and generalization in their response, emphasizing that robustness in this study is tied to resilience against violations of benchmark assumptions. This clarification addresses the original concern, though ensuring this explanation is explicitly included in the revised manuscript remains important.
2. The authors clarified that MedFuzz is designed to surface biases already present in the target model and does not introduce confusion into clinical decision-making itself. While this explanation addresses the primary concern, ensuring that the revised manuscript provides sufficient justification for the use of specific patient characteristics as perturbations will remain critical.
3. The authors acknowledged that the scale of perturbations could be further refined and suggested this as future work. Including a brief discussion in the revised manuscript about the implications of perturbation scale would strengthen this point.
4. The authors agreed to expand the analysis of CoT fidelity to include unsuccessful attacks in addition to successful ones. This addition should provide a more comprehensive baseline for evaluating the vulnerabilities identified by MedFuzz. Ensuring this analysis is effectively implemented in the revised manuscript will be crucial.

**Questions:**

1. It would be helpful to have specific examples illustrating the risks posed by the simplified assumptions in traditional benchmarks within clinical settings. For instance, if omitting certain patient characteristics or clinical contexts could lead to diagnostic errors, providing these examples would clarify the importance of this study for readers and highlight its relevance.

2. I am curious whether the patient characteristics (e.g., age, gender) and social bias information added as perturbations in MedFuzz genuinely act as confusion factors within actual clinical environments. These details often serve as crucial data points in clinical decision-making, so further explanation on how these elements were deemed appropriate as confusion-inducing factors would enhance the clinical validity of this study.

3. A clear explanation regarding the rationale for setting the perturbation iteration count to K=5 would be beneficial. For instance, do you have experimental results comparing the initial attack (K=1) with subsequent attacks (K=5) to illustrate how the LLM maintains performance with increasing perturbation levels? Such a comparison could provide a more reliable basis for evaluating the impact of iteration count on robustness in this study.

**Details Of Ethics Concerns:**

In the MedFuzz study, patient characteristics (PC) such as age, gender, race, and socioeconomic factors are added as perturbations to induce confusion in LLMs. One specific example presented by the authors is the use of “excessive hospital service usage by low-income patients.” This type of information could inadvertently reinforce social biases or perpetuate negative perceptions about certain demographic groups, rather than reflect clinical validity or fairness.

When such characteristics are introduced as confusion-inducing factors, there is a risk that essential background information—critical for accurate diagnosis and treatment—could lead to biased outcomes. Therefore, further clarification and evaluation are needed to ensure that MedFuzz’s inclusion of such data as perturbations aligns with clinical relevance and fairness, and to mitigate any potential reinforcement of harmful social biases in the model.

No further questions

---

> ### Author Response · Authors · 2024-11-26
> **Response to Reviewer 6sJS's feedback**
>
> We appreciate the reviewer’s detailed evaluation and thoughtful feedback on our manuscript. Below, we address each of the concerns and questions raised.
>
> ### **1. Definition of Robustness vs. Generalization**
> Robustness in the context of MedFuzz refers to the resilience of a model’s performance statistic (e.g., accuracy) when assumptions underlying the benchmark are violated in real-world settings. This includes maintaining performance when diagnostically irrelevant details are introduced. By contrast, generalization in statistics refers to the ability of a model to perform well on unseen data sampled from the same distribution as the training data.
>
> We will revise the manuscript to clarify this distinction and emphasize that robustness here is specifically tied to the benchmark’s assumptions and the model’s ability to handle clinically irrelevant or misleading details.
>
> ### **2. Patient Characteristics and Bias**
> We regret that we weren't more clear about how the use of patient characteristics (PC) in MedFuzz does not introduce or reinforce bias. Rather, it aims to surface biases already implicit in the target model. MedFuzz is a diagnostic tool to evaluate LLMs before they are deployed in clinical decision-making scenarios.
>
> Importantly, MedFuzz itself does not serve answers to questions in clinical settings—it evaluates the robustness of models that do. In that evaluation, it does not change or modify the target model. This distinction will be made clearer in the revised manuscript.
>
> ### **3. Scale of Perturbations**
> We did not constrain the proportion of added text during perturbations because, in our experience, the length of added text was still well within the length of the context windows for the target LLMs. We agree with the reviewer that analyzing how varying amounts of irrelevant information impact target model performance would be valuable. We will include this as a suggestion for future work.
>
> ### **4. Chain-of-Thought Fidelity**
> The CoT analysis focused on successful attaks to demonstrate that inspecting CoT explanations alone is insufficient to reveal the vulnerabilities surfaced by MedFuzz. We will expand the analysis to include unsuccessful attacks as well.
>
> ### **5. Examples of Benchmark Assumption Errors**
> The manuscript cites examples of errors that are not caught by traditional benchmark evaluation due to simplifying assumptions in those benchmarks. For example, we cite references showing GPT-3 demonstrating biases toward certain patient groups. We will expand on these examples in the revised manuscript to better illustrate the risks posed by such assumptions.
>
> ### **6. Ethical Concerns Regarding Bias**
> We address the ethical concerns raised by clarifying that MedFuzz is designed to surface biases in the target model, not to introduce or reinforce them. MedFuzz operates as an evaluation tool, diagnosing vulnerabilities in LLMs that may be deployed in clinical settings.
>
> We explicitly state that failure to surface such biases does not imply their absence. Furthermore, MedFuzz is not intended to answer medical questions but rather to assess the robustness of models that do. We will revise the manuscript to better highlight these points and allay concerns about bias reinforcement.
>
> ### **7. Perturbation Iteration Count \(K\)**
> The results for different values of \(K\) are shown in Figure 2. We demonstrate how performance changes as the number of perturbation iterations increases, providing empirical support for the choice of \(K=5\) as a practical balance between computational cost and perturbation effectiveness. We will ensure that this explanation is clearly referenced in the manuscript.
>
> ### **Revisions to the Manuscript**
> To address the reviewer’s feedback, we will:
> 1. Clarify the distinction between robustness and generalization, explicitly tying robustness to real-world violations of benchmark assumptions.
> 2. Emphasize that MedFuzz evaluates models to surface implicit biases, rather than introducing or reinforcing them.
> 3. Expand on examples of errors caused by traditional benchmark assumptions to strengthen the motivation for MedFuzz.
> 4. Expand the analysis of CoT fidelity to cover questions where attacks were unsuccessful, to establish a baseline for the analysis.
> 5. Ensure the ethical role of MedFuzz as an evaluation tool is clearly communicated.
> 6. Expand upon the discussion of \(K\) and iteration counts and how to select ideal values of \(K\).
>
> We appreciate the reviewer’s constructive feedback, which has helped us identify areas to strengthen the manuscript and address concerns. These revisions will further clarify MedFuzz’s methodology, ethical considerations, and contributions to LLM robustness evaluation.

---

> > ### Comment · Reviewer_6sJS · 2024-11-27
> > **MedFuzz: Exploring the Robustness of Large Language Models in Medical Question Answering**
> >
> > Thank you for your detailed and thoughtful responses to my feedback. I look forward to reviewing your revised manuscript, which I trust will sufficiently address the concerns raised, particularly regarding the distinction between robustness and generalization, the role of patient characteristics in MedFuzz, and the ethical considerations surrounding bias.

---

### Official Review · Reviewer_EcvC · 2024-10-31

**Soundness:** 3
**Presentation:** 3
**Contribution:** 2
**Rating:** 3
**Confidence:** 4

**Summary:**

The paper proposes an adversarial method for evaluating LLM performance on medical question-answering benchmarks to assess their robustness in real-world clinical settings. The idea is to automatically generate new question-answer pairs from the existing benchmark such that they still represent realistic scenarios (e.g., including additional patient information) but the answers remain the same. The experiment results demonstrate that various baseline LLMs can be tricked into providing incorrect answers.

**Strengths:**

* The idea of the paper is interesting -- existing medical QA datasets are fairly simplified and may not appropriately represent real-world clinical settings. Thus, there is a need to understand how safe LLM usage is for the medical domain via robustness analysis.
* The intuition for the adversarial biasing comes from medical domain understanding of the benchmark constructions.
* Authors benchmark 3 closed LLMS and 4 open-source, medically fine-tuned LLMs.

**Weaknesses:**

* One of the major claims of the method is that it will generate new questions that are semantically coherent and will not fool clinicians. However, there is no empirical proof that this is the case other than the analysis of a handful of case studies (one is presented in the main text). The prompt contains instructions for the attacker LLM it should not change the default answer but GPT-4 is not always guarenteed to follow the instructions or have all the correct medical knowledge appropriate.
* Is there a reason why general domain adversarial prompting wasn't shown to be sufficient? A few studies are listed in 2.2 (first sentence) but no preliminary studies or experimental studies are shown to support this.
* GPT-4 is chosen as the attacker LLM, but the question is why aren't other open-source models explored? In looking at OpenBIOLLM-70B performance, this also looks like a reasonable comparison to try and might even generate harder cases with less of the computation cost.
* One of the comments in the introduction was the that existing benchmarks are not challenging enough including reducing real-life clinical situations to canonical multiple choice questions. Is there a reason why only one dataset was included and it was a multiple-choice one?
* The statistical test is proposed to identify the significance of a successful attack using control fuzzes and to select the case studies, but what about the general distribution for the MedQA dataset? How stable is it broadly in identifying how significant a successful attack is? I understand this can be computationally intensive and costly but that also raises a bit of questions regarding the applicability of the method if it can't be done at scale.
* The presentation could have been improved to provide some intuition at the beginning with potentially a simpler case study where less was added to make the LLM response change. Similarly, some of the text is written in a less digestible format. For example, the introduction of the test statistic could be improved by introducing notation first and then how you might compute it to understand what the statistic is looking to capture.
* The citation format is incorrect, please use \citep instead of \cite as it detracts from readability.

**Questions:**

* Why was MedQA the only dataset used? There are a few other multiple choice medical QA ones liked MedMCQA, PubMedQA, and MMLU Clinical topics. Why MedQA?
* Why was only GPT-4 used as the attacker LLM? Seemingly there are other open source ones that have just as much medical knowledge especially looking at the fine-tuned example.
* The workflow for the Step 2 is quite a few iterative turns. Are they all necessary to generate grounded ones? Is this workflow generalizable to other LLMs? Or is it GPT-4 specific?

---

> ### Author Response · Authors · 2024-11-26
> **Response to Reviewer EcvC**
>
> We thank the reviewer for their detailed and thoughtful feedback on our manuscript. Below, we address each of the points raised and clarify our methodological choices. Firstly, we will correct the `\citep` citation format throughout the manuscript.
>
> ### **1. Empirical Validation of Semantically Coherent Fuzzes**
> Our qualitative evaluation of the fuzzed questions relies on feedback from medical expert users who review successful attacks and assess their plausibility. While this approach is effective in surfacing interesting cases, we recognize the need for more systematic and quantitative validation to empirically verify that clinicians would consistently provide correct answers to fuzzed questions. This limitation will be addressed in future work as part of broader medical expert evaluation efforts.
>
> ### **2. Use in Other Domains**
> The approach used in MedFuzz would apply in other domains. The approach relies on a domain expert who design the attacks and evaluate the results. The domain should also face serious robustness challenges in deploying in real-world settings. We chose medicine because of our experience with challenges in this domain.
>
> ### **3. Choice of GPT-4 as the Attacker LLM**
> We selected GPT-4 as the attacker LLM due to its exceptional performance on MedQA.  The attacker LLM must perform at least at a human level on the benchmark to effectively generate attacks that preserve the correct answer while introducing subtle, diagnostically irrelevant distractors. GPT-4 has also demonstrated performed well on theory of mind tests (Strachan et al., 2024), suggesting it would be good at generating ways to "trick" a test taker (the target LLM).
>
> We recognize the potential value of exploring fine-tuned open-source models like OpenBioLLM-70B as attackers. However, current fine-tuned models lack performance on this benchmark and demonstrated generalist reasoning abilities in other settings. In future work, we aim to investigate whether fine-tuning open-source models can achieve similar attacker capabilities at a lower computational cost.
>
> ### **4. Use of MedQA Dataset**
> We selected MedQA because it remained a challenging benchmark for state-of-the-art language models until GPT-4 and its direct competitors achieved near-human performance. To demonstrate MedFuzz’s value, the target LLM needed perform well enough on the benchmark to reveal meaningful vulnerabilities beyond just not understanding the questions.
>
> Expanding MedFuzz to other datasets like MedMCQA, PubMedQA, or MMLU Clinical Topics is an exciting direction for future work. The challenge with these datasets is that their variety in answer format and topic makes it challenging to identifying assumptions to violate that don't hold up in clinical settings. Relative to MedQA, they do not align as closely with our specific focus on robustness to real-world assumptions.
>
> ### **5. Scalability of the Statistical Test**
> The computational expense of the statistical test arises primarily from generating control fuzzes. For multiple-choice benchmarks, we recommend generating at least 30 control fuzzes per attack to ensure granularity in p-values that align with conventional significance thresholds. In future work, we plan to extend this methodology to open-ended answers by embedding generated responses and deriving p-values from the embeddings. We leave this to future work because it will require theoretical treatment as well as a much larger number of control fuzzes it will improve applicability to a wider range of benchmarks.
>
> ### **6. Iterative Workflow**
> The iterative workflow is not specific to GPT-4 and can be applied to other high-performing models like Claude Sonnet.
>
> Iterative turns are necessary to refine fuzzes, leveraging feedback from the target LLM to ensure attacks are semantically coherent and effective. While single-shot attacks are simpler, they often fail to exploit the nuanced vulnerabilities in advanced LLMs, as demonstrated by our initial experiments with single-turn methods (these were negative results will be added to the appendix for transparency).
>
> ### **7. Presentation and Intuition**
> We appreciate the reviewer’s suggestion to improve readability. In the revised manuscript, we will:
> - Add a simpler case study early in the text to illustrate the method.
> - Reorganize the introduction of the test statistic to introduce notation first, followed by an explanation of how it captures the significance of successful attacks.
>
> ### **Summary**
> We are grateful for the reviewer’s feedback and have outlined revisions to enhance the clarity, scalability, and rigor of our work. These include:
> 1. Adding negative results from single-shot attacks to the appendix.
> 2. Revising sections for improved readability and presentation.
> 3. Expanding the discussion of iterative workflows and generalizability to other datasets and models.
>
> We thank the reviewer for their thoughtful suggestions and are confident that these updates will strengthen the manuscript.

---

### Official Review · Reviewer_Dsnm · 2024-11-03

**Soundness:** 3
**Presentation:** 3
**Contribution:** 3
**Rating:** 6
**Confidence:** 3

**Summary:**

The paper proposes an automated red teaming approach to attack LLMs. They attempt this in the medical context by modifying medical Q&A datasets (specifically on MedQA), by violating assumptions that do not hold good in real life settings. The goal of MedFuzz is to make LLMs provide the wrong answer while ensuring that clinicians can still provide the right answer. The authors have identified a crucial problem with the evaluations of LLMs in the medical domain and provided a way to generate a more realistic dataset to aid subsequent LLM evaluation. The novelty lies in the proposed dataset from MedFuzz and the statistical evaluation used to check if the attack was successful.

**Strengths:**

•	Clarity: The paper is well written and easy to follow along. The authors have given adequate and clear examples at appropriate locations in the draft to aid readability. Good use of illustrations after consultation with domain experts (clinical collaborators in this case). The authors have also acknowledged the limitation of using contaminated training data.

•	Originality: The idea to use social biases a clever way to incorporate real life information into the MedQA dataset.

•	Quality: The evaluation involves the use of proprietary vs open source and general purpose vs domain specific models. The experiment settings for reproducibility like temperature have been provided. The approach should be easy enough to reproduce.

•	Significance: The authors have tackled a relevant problem that needs to be addressed, given the rapid pace of the domain.

**Weaknesses:**

•	In the case of MedQA dataset, the authors have identified a social bias which may be present in real life situations, which are removed in the original benchmark. It is unclear how easy it is to identify and exploit such peculiarities in other medical benchmarking datasets like MedMCQA[1], PubMedQA[2] etc.

•	The authors create the adversarial questions by an iterative multi-turn approach. Although the authors allude to the target LLM forgetting about previous Q&A attempts, would the approach be better validated if the evaluation is done in a single-turn manner?

•	The authors, in step 4, only validate the statistical significance of 4 individual interesting cases. How would this change if considered for all successful cases?

[1] Pal A, Umapathi LK, Sankarasubbu M. Medmcqa: A large-scale multi-subject multi-choice dataset for medical domain question answering. InConference on health, inference, and learning 2022 Apr 6 (pp. 248-260). PMLR.

[2] Jin Q, Dhingra B, Liu Z, Cohen WW, Lu X. Pubmedqa: A dataset for biomedical research question answering. arXiv preprint arXiv:1909.06146. 2019 Sep 13.

**Questions:**

•	The authors can clarify how their approach to adversarial attacks differs from the misinformation approach in [1].

•	The authors can clarify why unfaithfulness of generated responses is a crucial dimension to consider.

•	Section 2.2 Lines 104: The authors mention “two ways” in which MedFuzz differs from other adversarial ML approaches, though only one distinction is clear in the draft. I’m assuming the second way is the use of semantically coherent changes to the text. These few lines can probably be rephrased to add clarity.

•	The authors have conducted their experiments on the MedQA dataset and taken advantage of a constraint imposed in the curation of this dataset. The authors could potentially add broad guidelines to expand on the fuzzing idea for other medical datasets.

•	How can the authors ensure that the GPT-4 generated attack retains the same answer as the original QA pair being perturbed? Is there a possibility to evaluate this with the help of domain experts?

•	How is the value of K set in Algorithm 1? This can be elaborated on in the Appendix section.

•	Does the finding that LLM CoT does not mention the fuzzed information provide a way forward to identify adversarial inputs?

•	Another interesting avenue would be to examine how different kinds of LLMs perform when used as the attacking/ target LLM. For example, can a smaller model generate adversarial inputs faster than a larger model like GPT-4?

•	Minor Comment: Is line 10 a duplicate of line 11 in Algorithm 1?

[1] Han T, Nebelung S, Khader F, Wang T, Müller-Franzes G, Kuhl C, Försch S, Kleesiek J, Haarburger C, Bressem KK, Kather JN. Medical large language models are susceptible to targeted misinformation attacks. npj Digital Medicine. 2024 Oct 23;7(1):288.

**Details Of Ethics Concerns:**

NA. Authors have provided an ethics statement in the draft as well.

---

> ### Author Response · Authors · 2024-11-26
> **Response to Reviewer Dsnm**
>
> We appreciate the reviewer’s thoughtful and constructive feedback on our manuscript. Below, we address each of the points raised and clarify aspects of our approach.
>
> ### **1. Single-Turn Attacks**
> We initially explored single-shot attacks as a baseline approach. For example, with GPT-4 achieving 88.5% accuracy on the MedQA benchmark, we created several modified datasets that added diagnostically irrelevant patient characteristics. These datasets included patients characterized by varying socioeconomic statuses (e.g., affluent or low-income) and different racial or ethnic groups (Asian, Black, Hispanic, Native American, White), while excluding questions where race was clinically relevant. Across these datasets, no statistically significant change in accuracy was observed, indicating that such single-turn perturbations were too “easy” for advanced models like GPT-4. These findings inspired MedFuzz's multi-turn approach. We can include these negative results in the appendix to demonstrate the progression of our method.
>
> ### **2. Expanding Statistical Validation**
> We strongly adhere to the conventional statistical approach of having the end user evaluate interesting results and then using significance tests to validate that these findings contain signal. Expanding significance testing to a broader set of results would necessitate multiple comparisons corrections, which we leave for future work. Furthermore, ranking results based on p-values invites the risk of p-hacking, which we aim to avoid.
>
> ### **3. Faithfulness of Responses**
> The faithfulness analysis is not intended as a core contribution but rather as a supplementary finding to highlight that vulnerabilities revealed by MedFuzz cannot simply be detected by inspecting CoT explanations. We agree this distinction can be emphasized more clearly in the manuscript.
>
> ### **4. Comparison to Misinformation Attacks in Han et al. (2024)**
> MedFuzz fundamentally differs from the approach in Han et al. (2024). The attacks described in that work aim to poison target LLMs by injecting falsehoods during model updates, requiring access to gradients and training data. In contrast, MedFuzz does not poison, it *detects* “poison”, and does so without access to gradients or the data used for model updates.
>
> We will clarify this distinction in the manuscript to address the reviewer’s concern.
>
> ### **5. Guidelines for Expanding MedFuzz**
> MedFuzz’s approach is generalizable to any domain where benchmarks rely on performance statistics (e.g., accuracy) that are contingent on assumptions not robust in real-world settings. While our study focuses on medical datasets requiring clinical expertise, domain experts in other fields can evaluate MedFuzz outputs for their respective use cases.
>
> For medical benchmarks like MedMCQA and PubMedQA, the key challenge is identifying assumptions analogous to those violated in MedQA. We can provide broad guidelines for extending MedFuzz, such as focusing on domain-specific biases, assumptions, or oversights that simplify real-world complexity.
>
> ### **6. Ensuring Correct Answers in Fuzzed Questions**
> We rely on the attacker LLM’s high performance on MedQA to generate effective attacks while preserving the correct answer. Medically experienced users validate successful attacks by inspecting outputs and running significance tests. When the attacker fails to “fuzz” the question well, this is discovered during that human evaluation step. We plan for extensive human medical expert evaluation in future work.
>
> ### **7. Value of \(K\) in Algorithm 1**
> The ideal value of \(K\) (number of iterations) depends on the target model’s capabilities on a given benchmark. We will update the manuscript to suggest tuning \(K\) on a pilot subset of the data, increasing it incrementally until the marginal gains from additional iterations are no longer worth the computational expense.
>
> ### **8. Smaller Models as Attackers**
> We believe that the attacker model must have reached human-level performance on the benchmark to identify effective attacks. This ensures that the attacker LLM can leverage its understanding of the benchmark and the correct answer to generate meaningful perturbations. Smaller models effectiveness would be limited by their lower performance on the benchmark. Exploring this tradeoff is a promising direction for future work.
>
> ### **9. Redundancy in Algorithm 1**
> Thank you for pointing out the Algorithm 1, we will clarify this in the revised manuscript.
>
> ### **Summary**
> We appreciate the reviewer’s positive assessment of the manuscript’s clarity, originality, quality, and significance. We have provided clarifications and plan to incorporate additional results and updates in the revised manuscript, including:
> 1. Negative results from single-shot attacks in the appendix.
> 2. Broader guidelines for applying MedFuzz to other domains.
> 3. Refinements to Algorithm 1 and expanded discussion on parameter tuning.
>
> We thank the reviewer for their valuable feedback.

---

> > ### Comment · Reviewer_Dsnm · 2024-11-26
> >
> > Thank you for your detailed response! I shall wait for the revised paper to look at the edits made.

---

### Official Review · Reviewer_GdQb · 2024-11-04

**Soundness:** 2
**Presentation:** 2
**Contribution:** 2
**Rating:** 3
**Confidence:** 4

**Summary:**

This paper investigates the robustness of large language models in handling medical QA tasks by introducing a new evaluation method, MedFuzz. For each multiple-choice question in the original benchmarks, MedFuzz uses an LLM (referred to as the attacker LLM) to reformulate questions by adding patient characteristics that may introduce social bias without affecting the clinical decision-making process. If the target LLM answers correctly, the attacker LLM is prompted to generate additional distracting questions based on the target LLM’s feedback. Additionally, a non-parametric statistical significance test was developed by prompting the attacker LLM to create questions with patient characteristics that avoid social bias. Using this evaluation method, the authors tested seven LLMs and found a significant performance drop across all models. Moreover, they observed that when current LLMs answer incorrectly, they tend not to reference the added biased information, indicating inconsistency in faithfully adhering to the clinical decision-making process.

**Strengths:**

+ This paper examines the robustness of LLMs in the clinical decision-making process, a critical aspect of their application in the medical domain.

+ The evaluation results demonstrate that current LLMs lack robustness in the clinical decision-making process, offering valuable insights for the development of medical LLMs.

**Weaknesses:**

+ A major weakness of this paper is the faithfulness of the reformulated questions. The proposed MedFuzz method relies solely on prompt engineering with the attacker LLM (GPT-4) to modify original MedQA questions, making the attack process difficult to control. The attacker LLM may potentially alter critical information in the original questions, resulting in less reliable reformulated questions. The example in Section 3.1 also demonstrates that the attacker LLM added extensive information about the patient’s family medical history, consultation history, and medication history. These details are highly relevant in real clinical diagnosis and can significantly influence a doctor’s assessment of the patient’s condition.

+ Moreover, although the authors propose a non-parametric statistical significance test, they do not provide the full distribution of p-values across the MedQA benchmark. In line 485, they note that for the successful attacks they selected, the p-values are <1/30, 0.1, 0.16, 0.5, and 0.63. Here, the p-value represents the probability that a control fuzz is more challenging than the original fuzz. Therefore, cases with p-values of 0.5 and 0.63 suggest that the performance decline in the target LLM is due to the perturbations themselves, rather than social bias.

+ For the study of target LLM's faithfulness, it is important to also study the proportion of CoT that mentions the critical information in the original MedQA benchmark for comparison with the results provided in Figure 2B. Additionally, the authors should provide more information to help readers understand the specific process of this study. For example, how many cases were analyzed? Was the determination of whether fuzzed information was included made manually, or was an automated algorithm used?

**Questions:**

1. The authors need to provide further experiments and analyses to demonstrate the reliability of the questions generated by this method, such as incorporating the performance of human experts or introducing relevant methods for quality control of the questions in the methods section.

2. Also, more analysis of the evaluation results should be included. For example, what are the main types of errors introduced by attacks across different turns? Which specific diseases or problem types is the target LLM less robust against? By supplementing these analyses, further insights can be provided for the development of medical LLMs.

---

> ### Author Response · Authors · 2024-11-25
> **Response to Reviewer GdQb: Questions about P-value Distribution, Trends in Duccessful Attacks, and Human Evaluation**
>
> We thank the reviewer for their thorough evaluation and constructive feedback on our manuscript. Below, we address each point raised:
>
> ### **Faithfulness of Reformulated Questions**
> We acknowledge the concern about the reliance on the attacker LLM (GPT-4) to maintain the medically correct answer while generating fuzzed questions. In our approach, the attacker LLM is explicitly prompted to preserve the correct answer, which is provided during the fuzzing process. This ensures that the fuzzes remain anchored to the original question's intent. Furthermore, we rely on the attacker LLM’s demonstrated human-level accuracy on the benchmark as an assumption for generating high-quality fuzzes at scale.
>
> Given the large scale of MedFuzz experiments, manual quality assurance for every fuzzed question is infeasible. However, our workflow incorporates user inspection for particularly interesting or insightful cases, with the final judgment of whether an attack is “fair” being left to human reviewers. While this assumption introduces some dependence on the attacker LLM’s capabilities, we believe it is reasonable for achieving scalability.
>
> ### **Distribution of P-Values**
> To address the request for an impression of the p-value distribution, we ran an analysis on a run where GPT-4 was the target model, resulting in 85 successful attacks. Below are summary statistics of the p-values:
>
> - **Min:** 0.0, **5%:** 0.0, **25%:** 0.10, **Median:** 0.40, **Mean:** 0.408, **75%:** 0.63, **Max:** 1.0
>
> To explore trends, we categorized successful attacks into two groups: (1) significant attacks (\( p < 0.01 \)) and (0) insignificant attacks. We calculated an odds ratio for being in group 1 vs group 0. Analysis revealed that topics like “rash,” “substance abuse,” and “ultrasound” were more than twice as likely to fall into the significant group, while others like “HIV,” “breastfeeding,” and “chronic kidney disease” were also overrepresented. However, we recognize that calculating p-values for these odds ratios would conflate p-values used as thresholds, resulting in unsound statistical inference (i.e., p-hacking).
>
> A more robust approach, which we plan to explore in future work, would involve estimating the success probability for specific topics using repeated attacks on individual questions.
>
> ### **Evaluation of Chain of Thought (CoT) Faithfulness**
> The reviewer highlights an important point about the assessment of CoT faithfulness. In our study, we manually evaluated CoTs from successful attacks, focusing on whether the added fuzz content was explicitly referenced. This process was conducted by inspecting each CoT explanation and verifying its alignment with the fuzzed information that caused the incorrect response.
>
> ### **Human Performance Comparison and Quality Control**
> We recognize the value of including human medical experts to evaluate the quality of fuzzed questions. However, due to resource constraints, this was not feasible in the current study. We plan to include human evaluation in future work to provide an additional layer of validation for the attacker LLM’s performance and the robustness of fuzzed questions.
>
> Regarding quality control, we rely on the attacker LLM’s prompt-engineered constraints to ensure that generated fuzzes are medically plausible and consistent with the original question’s correct answer. This reliance on a high-performing LLM is a tradeoff we make to scale the MedFuzz method across a large dataset like MedQA.
>
> ### **Analysis of Errors and Robustness to Specific Problem Types**
> The reviewer’s request for a more granular analysis of errors and model vulnerabilities is well-taken. Our exploratory analysis of attack outcomes highlighted certain topics (e.g., “rash,” “substance abuse,” “ultrasound”) that appear more susceptible to significant attacks. However, as noted above, a more rigorous approach to topic-level success probability estimation is necessary for conclusive insights. We plan to develop a framework for repeated attacks on specific topics, allowing us to model robustness conditional on problem type.
>
> ### **Future Directions**
> The reviewer’s suggestions align with our broader vision for improving MedFuzz. Specifically, we aim to:
> 1. Incorporate human evaluation for assessing question quality and attack outcomes.
> 2. Develop Monte Carlo-based topic-specific estimates of probability of attack success to give insight into which topics are vulnerable.
>
> We believe these improvements will address the limitations of the current study and enhance the utility of MedFuzz for evaluating medical LLMs.
>
> **Summary**
> We appreciate the reviewer’s insightful feedback and have outlined both responses to the identified weaknesses and concrete steps for future work. While some limitations remain, we are confident that MedFuzz provides valuable insights into LLM robustness and look forward to building on this foundation.

---

### Meta-Review · Area_Chair_6xkD · 2024-12-15

**Metareview:**

In this paper, the authors propose MedFuzz, an LLM-based technique to provide challenging medical questions. The technique allows to test medical LLMs at scale on a multichoice QA dataset, by modifying factors deemed as 'irrelevant' to the final diagnosis.

While all reviewers found the approach interesting, significant concerns were raised in terms of the validity of the generated attacks, and whether these would be clinically plausible or indeed maintain the same response. Further concerns have been pointed out in terms of analyzing the effect of the choice of the LLM attacker, the irrelvant factors or the choice of K, or the choice of MedQA as the single dataset considered. Most of these are pointed to as future work by the authors, but I believe more depth is needed before publication. Therefore I recommend rejection.

**Additional Comments On Reviewer Discussion:**

The authors provided a response very late in the discussion period, which prevented revewers from engaging in depth. They also did not provide a revised version of the paper, which the reviewers would have wished to see.

---

### Decision · Program_Chairs · 2025-01-22

Reject